# Structural Identification of *Physalis alkekengi* L. Polysaccharides

**DOI:** 10.3390/molecules30040949

**Published:** 2025-02-18

**Authors:** Yun Zhang, Xuan Wen, Neng Xu, Hongyan Fu, Ge Lv, Wenjie Yu, Lina Wei, Lin Zhao

**Affiliations:** 1College of Food Engineering, Heilongjiang East University, Harbin 150066, China; 15536278415@163.com (X.W.); 15715737669@163.com (N.X.); hongyan770419@163.com (H.F.); gracelvge@163.com (G.L.); yuwenjiedf@163.com (W.Y.); ww8381only@126.com (L.W.); 2Quality & Safety Institute of Agricultural Products, Heilongjiang Academy of Agricultural Sciences, Harbin 150086, China; zhaolin78@126.com

**Keywords:** *Physalis alkekengi* L., polysaccharide, ion chromatography, methylation, nuclear magnetic resonance spectrum analysis, structural identification

## Abstract

*Physalis alkekengi* L. fruit polysaccharides can reduce blood sugar, regulate blood lipids, and improve intestinal flora structure. However, the specific polysaccharide components exerting these effects are unclear. In this study, we extracted, separated, purified, and characterized the *P. alkekengi* polysaccharides Phy-1a, Phy-1b, and Phy-1c. Ion chromatography showed that Phy-1b was mainly composed of rhamnose, arabinose, galactose, glucose, and xylose at a molar ratio of 3.0:19.8:47.5:20.9:8.8, and Phy-1c was composed of rhamnose, arabinose, galactose, glucose, xylose, mannose, ribose Galactosamine hydrochloride and Glucosamine hydrochloride at a molar ratio of 10.4:7.9:22.8:30.5:4.6:4.4:19.4:3.9:5.8. Neither of these polysaccharides contained uronic acid, indicating their neutral property. Methylation analysis and nuclear magnetic resonance spectroscopy showed that Phy-1b was mainly composed of terminal sugars (1-Araf); 1,5-Araf; 1,4-Xylp; 1-Glcp; 2,4-Rhap; 1,3-Glcp; 1,4-Galp; 1,4-Glcp; 1,3-Galp; 1,6-Glcp; 1,3,6-Glcp; and 1,4,6-Galp at a molar ratio of 5.2:7.1:7.8:13.7:6.3:11.2:7.0:16.3:7.4:6.0:6.8:5.3, with the main chain being →2)-α-L-Rhap-(1→4)-β-d-Galp-(1→4)-β-d-Galp-(1→[3)-β-d-Glcp-(1]2→3)-β-d-Glcp-(1→[4)-β-d-Glcp-(1]2→ and the branched chains being β-L-Araf-(1→5)-β-L-Araf-(1→, β-d-Glcp-(1→4)-β-d-Xylp-(1→ 3)-β-d-Galp-(1→, and β-d-Glcp-(1→6)-β-d-Glcp-(1→. The three fragments, respectively, pass through the O-4 key of →2,4)-α-l-Rhap-(1→, O-6 key of →4,6)-β-d-Galp-(1→, and O-6 of →3,6)-β-d-Glcp-(1→ connected to the main chain. These results provide a reference for enhancing the utilization value of *P. alkekengi* resources to promote its high-value and efficient processing.

## 1. Introduction

*Physalis alkekengi* L., also known as “Golden Lantern” and “Red Mushroom Niang”, is a perennial plant species in the genus *Physalis* of the family Solanaceae, which is cultivated in many areas of the world. At present, there are approximately 120 recognized species of *Physalis*, most of which are distributed in temperate regions [1]. After maturity, the calyx becomes yellow, orange, or red, and the fruit can be red, orange, or yellow. *Physalis* has a long planting history in China, Japan, and Europe [2], with five species/varieties currently widely distributed in China: Physalis, Kuteng, *Physalis alkekengi* L., Nymphaea, and *Physalis pubescens* [3]. Among them, *P. alkekengi* is mainly distributed in the northeast, northwest, and north of China, which are mostly present as wild resources. However, the planting amount is relatively small, and the cultivation of *P. alkekengi* in China is more extensive in the northeast [4]. The pulp is sweet, slightly sour, and cold in nature and belongs to the lung meridian in traditional Chinese medicine. *P. alkekengi* also has traditional medicinal functions of clearing away heat and toxic materials, relieving sore throat and resolving phlegm, and promoting diuresis and stranguria [5]. Studies have shown that *P. alkekengi* is rich in nutrients, including alkaloids, polysaccharides, flavonoids, sterols, carotenoids, amino acids, and organic acids [6]. Polysaccharides are the main components extracted from the fruit of *alkekengi*, which has medicinal activities of immunomodulation, antioxidation, lowering blood sugar and blood lipids, and regulating the intestinal flora [7,8].

Studies have proved that Guichang kiwifruit polysaccharides [9] have in vitro antioxidant effects, Wang et al. [10] proved that astragalus and ginseng polysaccharides have immunomodulatory effects on weaned piglets, Pei et al. [11] showed that Morchella polysaccharides can act as a good prebiotic to play a probiotic role. Zhou et al. [12] found that significant immunomodulatory activity was found in RAW264.7 cells. Ge et al. [13] found that acidic plasma polysaccharides can function as an anti-ageing agent by improving the vigour of relevant antioxidant enzymes in the body and can be less fatiguing and speed up the fatigue to subside. Antioxidant capacity affects diabetes mellitus; elevated levels of free radicals attack insulin receptors, leading to insulin resistance, and it has been found that diabetic patients have a reduced ability to scavenge free radicals, which leads to oxidative damage, thus worsening diabetes mellitus. Yang [14] extracted polysaccharides from Porogonella foetidis using a high-pressure ultrasound-assisted method. And its structure analysis and bioactivity analysis results found that the polysaccharide of Porcupine alga has good antioxidant activity, and it has good hypoglycemic effect.

With scholars’ in-depth research on the mechanism of hypoglycaemic effects of polysaccharides, the regulation of glucose metabolism by polysaccharides and the effect on insulin was found to be one of the important pathways for polysaccharides to regulate glucose metabolism. Tang et al. [15] found that Astragalus polysaccharide APS improved glucose and lipid metabolism in T2DM rats, and the pathway of action may be through protecting pancreatic β-cells from injury, thus increasing insulin secretion. Wang et al. [16]. studied gibberellic acid polysaccharide (GPP) and found that GPP could increase the activity of α-glucosidase, promote the metabolic uptake of glucose, and thus improve blood glucose in diabetic mice. Zhang et al. [17] found that the blood glucose level of DM mice gavaged with apricot mushroom polysaccharides was significantly reduced, and the dietary intake of drinking water in the DM model after gavaging apricot mushroom polysaccharides showed a tendency to return to the level of healthy mice. Zou et al. [18] purified and structurally analyzed the fruits of Lycium barbarum and found that this polysaccharide is an acidic polysaccharide consisting of xylose, galactose, rhamnose, arabinose, and mannose, while the polysaccharide was able to inhibit HepG2 cells thereby improving insulin resistance. Zhang et al. [19] investigated the hypoglycaemic efficacy of hairy plasma polysaccharides in diabetic (DM) mice. After 8 weeks of intervention with hairy plasma polysaccharides, the glucose tolerance of DM mice was improved, resulting in a reduction in fasting blood glucose levels.

Li et al. [20] found that tea polysaccharides significantly reduced fasting blood glucose as well as total cholesterol, total triglycerides, LDL cholesterol and free fatty acid levels in type 2 diabetic rats. Through metabolomics analysis, the researchers found that tea polysaccharides intervened to affect the production of amino acids and other related metabolites. Correlation analysis between metabolites, gut microbes and hypoglycemic parameters indicated that tea polysaccharides could exert their hypoglycemic and hypolipidemic effects in T2D by modulating gut microbes and improving host metabolism. Li et al. found that shiitake mushroom polysaccharides could promote the proliferation of T-lymphocytes, relieve tumour suppression of the host’s immune system, and enhance anti-tumour effects [21].

Liujing et al. [22] found that Schisandra chinensis polysaccharide has remarkable antioxidant capacity, which could alleviate oxidative damage, protect cells from free radicals, and maintain the normal function of cells under oxidative stress, thereby playing a potential role in preventing and treating diseases related to oxidative stress. Among the monosaccharide components of Schisandra chinensis polysaccharide, the contents of rhamnose, arabinose, and galactose were found to be closely related to the observed antioxidant effect. In recent years, with the development and application of modern analytical instruments, researchers have gained a clearer understanding of the structure of polysaccharides [23,24]. *P. alkekengi* L. polysaccharide was reported to exert hypoglycemic and antioxidant effects [25]. We recently found that *P. alkekengi* L. fruit polysaccharide can reduce blood sugar, regulate blood lipids, and improve intestinal flora structure of type 2 diabetes mellitus model mice (unpublished data). However, there are relatively few studies on the hypoglycemic effect of *P. alkekengi* polysaccharides. Moreover, research on the structure of *P. alkekengi* polysaccharides has mostly revealed only a preliminary characterization. Therefore, to best exploit the considerable development and utilization value of this plant [26], in this study, we systematically identified the structure of *P. alkekengi* L. polysaccharides, which will facilitate further research to confirm their functions.

## 2. Results

### 2.1. Gel Purification Results

The gel column chromatography purification map of Phy-1, the polysaccharide extracted from *P. alkekengi*, is shown in Figure 1. The symmetrical part of the first peak was collected in 36–48 min as component a; Phy-1a was obtained by rotary evaporation concentration, dialysis, and freeze-drying, with a yield of 9.1%. The symmetrical part of the second peak was collected in 70–80 min as component b, which was concentrated by rotary evaporation, dialyzing, and freeze-drying to obtain Phy-1b with a yield of 3.1%. The symmetrical part of the third peak was collected in 89–111 min as component c, which was concentrated by rotary evaporation, dialyzed, and freeze-dried to obtain Phy-1c with a yield of 8.2%.

### 2.2. Relative Molecular Mass of Polysaccharide Components

The molecular weight of each component of purified *P. alkekengi* polysaccharide was determined by high-performance liquid chromatography to verify the purity. The chromatograms are shown in Figure 2, and the calibration curve equations are as follows:lgMp-RT (peak molecular weight): Y = −0.1817X + 11.616 (R^2^ = 0.9955);lgMw-RT (weight average molecular weight): Y = −0.1936X + 12.171 (R^2^ = 0.9933);lgMn-RT (number average molecular weight): Y = −0.1794X + 11.453 (R^2^ = 0.9911).

According to the above equations, the peak molecular weight (Mp), weight average molecular weight (Mw), and number average molecular weight (Mn) of Phy-1a were 46 kDa, 59 kDa, and 39 kDa, respectively; those of Phy-1b were 8.6 kDa, 9.8 kDa, and 7.4 kDa, respectively; and those of Phy-1c were 8.7 kDa, 9.8 kDa, and 7.4 kDa, respectively. Among the three purified components, Phy-1b and Phy-1c exhibited a single symmetrical peak, and the peak area ratio was 100%, which indicated that the polysaccharide samples of these two components had high purity and could be used for subsequent structural tests.

### 2.3. Monosaccharide Compositions of Polysaccharide Components

Figure 3 shows the ion chromatograms of the mixed standard, hydrolyzed Phy-1b, and Phy-1c. The peak retention times of the two components were compared with that of the mixed standard to determine and analyze the monosaccharide composition. As shown in Table 1, Phy-1b was composed of rhamnose, arabinose, galactose, glucose, and xylose, with a molar ratio (according to the peak area calculation) of 3.0:19.8;47.5:20.9:8.8, demonstrating that galactose and glucose are the main monosaccharides of Phy-1b, which may be part of the main chain. Phy-1c is composed of rhamnose, arabinose, galactose, glucose, xylose, mannose, and ribose with a molar ratio of 10.4:7.9:22.8:30.5:4.6:4.4:19.4, again indicating that galactose and glucose are the main monosaccharides.

Different monosaccharide compositions of polysaccharides may lead to different biological activities; therefore, it is of great significance to analyze the monosaccharide composition of polysaccharides in *P. alkekengi*. Although the monosaccharide molar ratios of Phy-1b and Phy-1c were slightly different, their components were largely similar, comprising rhamnose, arabinose, galactose, glucose, and xylose, and no uronic acid component was detected in either fraction. This indicated that Phy-1b and Phy-1c are neutral polysaccharides and were thus suitable for methylation analysis.

### 2.4. Methylation Analysis Results

The types and proportions of glycosidic bonds of Phy-1b were determined by methylation analysis. Methylated sugar alcohol acetate (PMAA) of Phy-1b was obtained by methylation, hydrolysis, and acetylation of the sample, and its total ion flow diagram obtained by GC-MS is shown in Figure 4.

Comparison of the corresponding mass spectrum of PMAA peaks with the CCRC Spectral Database for PMAA of the University of Georgia, chemical data of the China Academy of Sciences, and related literature [27,28,29,30,31,32,33,34,35] revealed the glycosidic bond type and molar ratio of *P. alkekengi* polysaccharide Phy-1b. As shown from Table 2 below, the peak of PMAA of Phy-1b was identified as 2,3,5-Me3-Araf; 2,3-Me2-Araf; 2,3-Me2-Xylp; 2,3,4,6-Me4-Glcp; 3-Me1-Rhap; 2,4,6-Me3-Glcp; 2,3,6-Me3-Galp; 2,3,6-Me3-Glcp; 2,4,6-Me3-Galp; 2,3,4-Me3-Glcp; 2,4-Me2-Glcp; and 2,3-Me2-Galp. This analysis demonstrated that the glycosidic bond connection mode of Phy-1b is complicated, including 12 different glycosidic bond types. Among them, arabinose exists as a terminal sugar 1-Araf and 1,5-Araf; xylose exists as the 1,4-Xylp type; glucose is present in five types (the terminal sugar 1-Glcp; 1,3-Glcp; 1,4-Glcp; 1,6-Glcp; and 1,3,6-Glcp); rhamnose exists as the 1,2,4-Rhap type; and galactose is present in three types (1,3-Galp; 1,4-Galp; and 1,4,6-Galp). (See Table 2 below). The results of this analysis were consistent with the monosaccharide composition determined for *P. alkekengi* polysaccharide Phy-1b.

### 2.5. NMR Structural Characterization of Phy-1b

Figure 5 shows the one-dimensional NMR spectrum of Phy-1b. The chemical shift δ of anomeric hydrogen in the hydrogen spectrum (Figure 5a) was in the range of 3.0–6.0 ppm. In the hydrogen spectrum, the proton signal of the sugar ring is concentrated in the range of 3.2–4.0 ppm, and the signal peak of the chemical shift δ of the main terminal matrix sub-peak is concentrated in the range of 4.3–6.0 ppm. Figure 5b shows the carbon spectrum, demonstrating signal changes in the range of 60–120 ppm. From the carbon spectrum, it can be seen that the chemical shift δ of the main heterocarbon is mainly distributed between 93 and 180 ppm and the main signal peaks are distributed in the region of 18 to 180 ppm.

The HSQC chart in Figure 5c demonstrates that the chemical shift δ of the heterocarbon is 108.88 ppm, and the corresponding heterohydrogen signal δ is 5.01. According to the COSY chart in Figure 5d, the δ signal of H1-2 is 5.05/4.07, that of H2-3 is 4.07/3.94, that of H3-4 is 3.94/4.19, and that of H4-5a is 4.19/3.82, resulting in H1, H2, H3, H4, and H5a are 5.05, 4.07, 3.94, 4.19 and 3.82. The corresponding δ values of C1–C5 are 108.81, 82.18, 78.12, 83.68, and 68.27, respectively. Therefore, the signal belongs to the glycosidic bond →5)-β-L-Araf-(1→. The HSQC map in Figure 5d further shows that the δ of the heterocarbon is 108.61, and the corresponding heterohydrogen signal is δ5.17. According to the COSY diagram in Figure 5d, the δ signal of H1-2 is 5.17/4.13, that of H2-3 is 4.13/3.87, that of H3-4 is 3.87/4.05, and that of H4-5a is 4.05/3.75, thereby H1, H2, H3, H4, and H4 are inferred. The corresponding C1–C5 values are 110.66, 82.81, 78.27, 85.11, and 62.64, respectively. Therefore, the signal should be attributed to the glycosidic bond β-L-Araf-(1→.

Similarly, by combining the HMBC pattern in Figure 5e and the NOESY pattern in Figure 5f, all glycosidic bond signals could be classified, which are summarized in Table 3.

Therefore, when analyzing the branched chain, Figure 5f shows that the anomeric hydrogen of the glycoside bond β-L-Araf-(1→5)-β-L-Araf-(1→5)-α-l-Araf has a related signal peak, indicating the existence of β-L-Araf-(1→5). In the HMBC map, the heterohydrogen of the glycosidic bond →5)-β-L-Araf-(1→) has a signal peak with C4 of →2,4-α-l-Rhap-(1→), indicating the existence of →5)-β-L-Araf-(1→2,4)-α-L.

In addition, the glycosidic bond β-D-Glcp-(1→ isohead hydrogen has a correlated signal peak with its →4)-β-D-Xylp-(1→ H4, suggesting the presence of a β-D-Glcp-(1→4)-β-D-Xylp-(1→ linkage. The heterohydrogen of the glycosidic bond →4)-β-d-Xylp-(1→) has a signal peak with H3 of →3)- β-d-Galp-(1→). This indicates that there is a connection mode of→4)-β-d-Xylp-(1→3)-β-d-Galp-(1→). The heterohydrogen of glycosidic bond →3)- β-d-Galp-(1→) has a signal peak with H3 of →4,6)-β-d-Galp-(1→), demonstrating a connection mode of →3)-β-d-Galp-(1→4,6)-β-d-Galp-(1→). The heteroterminal hydrogen of β-d-Glcp-(1→) and →6)-β-d-Glcp-(1→) have related signal peaks, demonstrating a connection mode of β-d-Glcp-(1→6)-β-d-Glcp-(1→). The heterohydrogen of glycosidic bond →6)-β-d-Glcp-(1→) has a signal peak with its H6 of →3,6)-β-d-Glcp-(1→), demonstrating a connection mode of β-d-Glcp-(1→3,6)-β-d-Glcp-(1→).

For the main chain, in the HMBC spectrum, the heterohydrogen of →2,4)-α-l-Rhap-(1→) and C4 of →4)-β-d-Galp-(1→) have related signal peaks, indicating the existence of →2,4)-α-l-Rhap-(1→4)-β-d. In the HMBC spectrum, the heterohydrogen of →4)-β-d-Galp-(1→) has a signal peak with C4 of →4,6)-β-d-Galp-(1→), which indicates the existence of →4)-β-d-Galp-(1→4,6)-β-d. In NOESY analysis, the heterohydrogen of →4,6)-β-d-Galp-(1→) has a correlation signal peak with H3 of →3)-β-d-Glcp-(1→), indicating the existence of →4,6)-β-d-Galp-(1→3)-β-. In the NOESY map, the heterohydrogen of →3)-β-d-Glcp-(1→) has a correlation signal peak with its own H3, which indicates the existence of →3)-β-d-Glcp-(1 →. The heterohydrogen of →3)-β-d-Glcp-(1→) has a signal peak with H3 of →3,6)-β-d-Glcp-(1→), indicating the existence of →3)-β-d-Glcp-(1→3,6)-β-d-Glcp. In the NOESY map, the heterohydrogen of →3,6)-β-d-Glcp-(1→) has a correlation signal peak with H4 of →4)-β-d-Glcp-(1→), indicating the existence of →3,6)-β-d-Glcp-(1 → 4)-β-.

Combining the results of the monosaccharide composition analysis and methylation analysis, it can be concluded that the main chain of *P. alkekengi* L. polysaccharide Phy-1b is →2)-α-l-Rhap-(1→4)-β-d-Galp-(1→4), because of its low content of other monosaccharides and a primary composition of galactose, glucose, rhamnose, xylose, and arabinose. The branched chains are β-L-Araf-(1→5)-β-L-Araf-(1→, β-d-Glcp-(1→4)-β-d-Xylp-(1→3)-β-d-Galp-(1→, and β-d-Glcp-(1→6)-β-d-Glcp-(1→; these three fragments, respectively, pass through O-4 key of →2,4)-α-l-Rhap-(1→, O-6 key of →4,6)-β-d-Galp-(1→, and O-6 of →3,6)-β-d-Glcp-(1→ connected to the main chain (Figure 6).

## 3. Discussion

Polysaccharides are polymeric carbohydrates composed of glycosidic bonds and at least 10 monosaccharides. The molecular weight of polysaccharides can range from several thousand to millions of daltons (Da), with characteristics of different structures, various types, and broad activities [36]. Accumulating studies have confirmed that the biological activity of a polysaccharide is influenced by its spatial conformation, glycosidic bond type, monosaccharide composition, and branching degree [37]. For example, the substantial hypoglycemic effect of the *Pleurotus ostreatus* mycelium polysaccharide A1-MPS may be due to its β-glucosidic pyranose [38]. The β-linked mannose residue or the α-glycosidic bond at the end of the branched chain of *Dioscorea opposita* polysaccharide can trigger an innate immune response, while the β-(1→3) glycosidic bond in the main chain can regulate the polysaccharide to exert its immune activity, ultimately exerting anti-tumour, anti-inflammatory, and immunomodulatory effects [39]. We found that the *P. alkekengi* polysaccharide Phy-1b includes an α-glycosidic bond configuration and β-glycosidic bond configuration, with the latter being the primary configuration type. This suggests that the unique α- and β-configurations of Phy-1b contribute to the previously observed hypoglycemic activity of *P. alkekengi* polysaccharide. The polysaccharide extracted from the *Grifola frondosa* fruiting body was identified as a pyranose with an α-glucosidic bond structure, which exhibited potent immune activity [40].

## 4. Materials and Methods

### 4.1. Chemicals and Reagents

Methyl iodide and dimethyl sulfoxide were obtained in analytical-grade form from Adamas Reagent Co., Ltd. (Shanghai, China). Sodium hydroxide, with superior-grade purity, was obtained from Afa Aisha Chemical Co., Ltd. (Shanghai, China). Analytical-grade acetic anhydride, acetic acid, and methanol were obtained from Sinopharm Group Chemical Reagents Co., Ltd. (Shanghai, China). Dextra standard 3,693,000 (≥99% purity) was obtained from Polymer Standards Company (Mainz, Germany). Trifluoroacetic acid (analytical purity) and sodium acetate (excellent purity) were obtained from Thermo Fisher Scientific Co., Ltd. (Waltham, MA, USA). Ethyl acetate (analytical grade) was obtained from Shanghai Wokai Chemical Reagent Co., Ltd. (Shanghai, China). Dextran standard 1152 (99% pure) was obtained from Shanghai Yuanye Biotechnology Co., Ltd. (Shanghai, China). Sodium chloride (analytical grade) was obtained from Tianjin Guangfu Technology Development Co., Ltd. (Tianjin, China). Sodium borohydride, perchloric acid, heavy water, deuterated acetone (all analytical grade), and dextran standards 5000 (≥99%), 11,600 (99%), 23,800 (≥97%), 48,600 (≥99%), 80,900 (>98%), 148,000 (>98%), 273,000 (98%), 409,800 (>98%), and 667,800 (>98%) were obtained from Sigma–Aldrich Reagent Company (Darmstadt, Germany). Dextran gel, SUGAR-BRT-103, a methylation kit, and analytical-grade mannose, rhamnose, galacturonic acid, galactose, glucose, glucuronic acid, arabinose, xylose, fucose, d-fructose, and d-ribose were obtained from Yangzhou Borui Sugar Biotechnology Co., Ltd. (Yangzhou, China).

### 4.2. Instrumentation and Equipment

The RI-502SHODEX refractive index detector was obtained from Showa Electric Co., Ltd., Tokyo, Japan. BSZ-100 Automatic Collector and the BRT-GS gel purification system were purchased from Yangzhou Borui Sugar Biotechnology Co., Ltd. The FDU-1100 vacuum freeze-dryer was from Tokyo Physical and Chemical Co., Ltd., Chuo-ku, Japan. The LC-10A high-performance liquid chromatograph and RI-10A differential detector were from Shimadzu Company, Kyoto, Japan. The ICS5000 ion chromatograph was obtained from Thermo Fisher Scientific Co., Ltd. The UGC-24M nitrogen blower was from Guangzhou Lichen Technology Co., Ltd. (Guangzhou, China). The 6890-5973 gas chromatography–mass spectrometry (GS-MS) system was from Agilent Technology Co., Ltd. (Tokyo, Japan). The AVANCE III HD 600 MHz nuclear magnetic resonance (NMR) spectrometer was from Swiss Brooke Instruments Co., Ltd. (Tokyo, Japan).

### 4.3. Gel Chromatography Purification of Physalis alkekengi Polysaccharides

*Physalis alkekengi* L. was obtained from Shang Cao Tang Hong Gu Niang Foods limited company (Hong Kong, China). The DEAE Sepharose Fast Flow anion exchange chromatography packing was soaked in 0.5 mol/L hydrochloric acid for 1 h, and any impurities were washed away and eluted with 4–5 times distilled water to achieve neutrality. The flow rate was adjusted to 5 mL/min and balanced with distilled water for 2 h. The sample was dissolved in distilled water, heated, swirled, and centrifuged at 12,000 rpm, and the supernatant was collected. The flow rate was then adjusted to 15 mL/min, and the linear velocity was 4.54 cm/s. The sample was eluted with distilled water, followed by four groups of solvents: three column volumes of water, 0.2 M NaCl, 0.5 M NaCl, and 1.0 M NaCl. The phenol-sulfuric acid method was used for detection tracking at 490 nm on a Spectrophotometer. The components were identified according to the resulting peak shape; each component was collected, concentrated, dialyzed in a 3500-Da dialysis bag, and freeze-dried. The obtained components (polysaccharides) were named Phy-1. After weighing to an appropriate amount, Phy-1 was dissolved with the mobile phase, centrifuged at 12,000 rpm for 10 min, and the supernatant was collected as the sample. Phy-1 was then purified by an automatic gel purification system, combined with online detection and collection by a differential detector, and symmetrical peaks were determined. The collected liquid was dialyzed, concentrated by a rotary evaporator, and freeze-dried to obtain separated polysaccharides named Phy-1a, Phy-1b, and Phy-1c, respectively. After purification on a gel column, the respective yields of the three polysaccharides were calculated [27].

### 4.4. Determination of Relative Molecular Weight

The relative molecular weights of the purified polysaccharide components Phy-1a, Phy-1b, and Phy-1c from *P. alkekengi* were determined by gel chromatography with a DionexCarbopacTMPA20 (3 mm × 150 mm) column; mobile phase A was H_2_O, and mobile phase B was 15 mm NaOHC, 15 nm NaOH, and 100 mM NaOAC. The flow rate was 0.3 mL/min, the sample volume was 5 μL, the column temperature was 30 °C, and an electrochemical detector was used. The chromatogram was obtained by injection, and the standard curve was constructed with polyethylene glycol as the standard sample. The retention time of polysaccharide samples was measured after injection and the relative molecular weight was calculated according to the curve equation [27].

### 4.5. Analysis of Monosaccharide Composition

The monosaccharide composition of *P. alkekengi* polysaccharides was analyzed by ion chromatography as described previously [27,28].

### 4.6. Methylation Analysis

The neutral polysaccharide Phy-1b was subject to methylation, hydrolysis, and acetylation for further analysis with GC-MS and compared with the standard mass spectrometry library. The polysaccharide sample was weighed (5–15 mg) and placed in a glass reaction bottle with 1 mL of anhydrous dimethyl sulfoxide. Methylating reagent A solution was immediately added, the bottle was sealed, and the solution was dissolved under the action of ultrasound. Methylating reagent B solution was then added and left to react for 60 min in a magnetic stirring water bath at 30 °C. Finally, 2 mL of ultrapure water was added to the mixture to terminate the methylation reaction. The methylated polysaccharide was mixed with 1 mL of 2 M trifluoroacetic acid for hydrolysis for 90 min and evaporated with a rotary evaporator. Subsequently, 2 mL of double-distilled water was added to the residue, reduced with 60 mg of sodium borohydride for 8 h, glacial acetic acid was added for neutralization, and the sample was rotary-steamed and dried in an oven at 101 °C. The dried sample was acetylated with the addition of 1 mL of acetic anhydride, heated at 100 °C for 1 h, and then cooled. Three millilitres of toluene was added, and the sample was concentrated and evaporated under reduced pressure; this step was repeated four to five times to remove excess acetic anhydride. The acetylated product was dissolved in 3 mL CH_2_Cl_2_ and transferred to a separatory funnel. After adding a small amount of distilled water, the solution was fully shaken, and the upper aqueous layer was removed; this step was repeated four times. The CH_2_Cl_2_ layer was dried with an appropriate amount of anhydrous sodium sulphate at a constant volume of 10 mL and placed into a liquid-phase vial. Finally, the Shimadzu GCMS-QP 2010 GC-MS system was used to determine the acetylated products with an RXI-5 SIL MS column (30 m × 0.25 mm; 0.25 μm). The programmed heating conditions were as follows: the initial temperature was 120 °C, which was increased at 3 °C/min to 250 °C/min and then maintained for 5 min. The inlet temperature was 250 °C, the detector temperature was 250 °C/min, the carrier gas was helium, and the flow rate was 1 mL/min [27].

### 4.7. NMR Spectrum Analysis

Phy-1b (50 mg) purified from *P. alkekengi* polysaccharide was weighed, dissolved in 0.5 mL of heavy water, and freeze-dried. The freeze-dried powder was dissolved in 0.5 mL of heavy water again, and the freeze-drying was continued; this process was repeated to fully exchange the active hydrogen. After dissolving the sample in 0.5 mL of heavy water, the 1H-NMR, 13C-NMR, and one-dimensional and two-dimensional DEPT135 spectra were measured on an NMR instrument at 600 MHz at room temperature (25 °C) [27].

## 5. Conclusions

Phy-1b was obtained from *Physalis alkekengi* L. polysaccharides by extraction, separation, and purification. Ion chromatography showed that Phy-1b was mainly composed of rhamnose, arabinose, galactose, glucose, and xylose, with a molar ratio of 3.0:19.8:47.5:20.9:8.8. Methylation analysis showed that Phy-1b was mainly composed of terminal sugars (1-Araf); 1,5-Araf; 1,4-Xylp; 1-Glcp; 1,3-Glcp; 1,4-Glcp; 1,6-Glcp; 1,3,6-Glcp; 1,2,4-Rhap; 1,3-Galp; 1,4-Galp; and 1,4,6-Galp with a relative molar ratio of 5.2:7.1:7.8:13.7:11.2:16.3:6.0:6.8:6.3:7.4:7.0:5.3. The results of NMR spectrum analysis showed that the main chain of the *P. alkekengi* polysaccharide Phy-1b is →2)-α-l-Rhap-(1→4)-β-d-Galp-(1→4)-β-d-Galp-(1→[3)-β-d-Glcp-(1]2→3)-β-d-Glcp-(1→[4)-β-d-Glcp-(1]2→ and the branched chains are β-L-Araf-(1→5)-β-L-Araf-(1→, β-d-Glcp-(1→4)-β-d-Xylp-(1→ 3)-β-d-Galp-(1→, and β-d-Glcp-(1→6)-β-d-Glcp-(1→. The three fragments, respectively, pass through O-4 key of →2,4)-α-l-Rhap-(1→, O-6 key of →4,6)-β-d-Galp-(1→, and O-6 of →3,6)-β-d-Glcp-(1→ connected to the main chain.

## Figures and Tables

**Figure 1 molecules-30-00949-f001:**
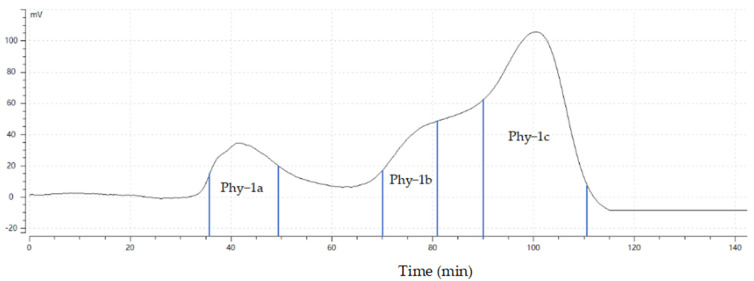
Gel purification map of *P. alkekengi* polysaccharide.

**Figure 2 molecules-30-00949-f002:**
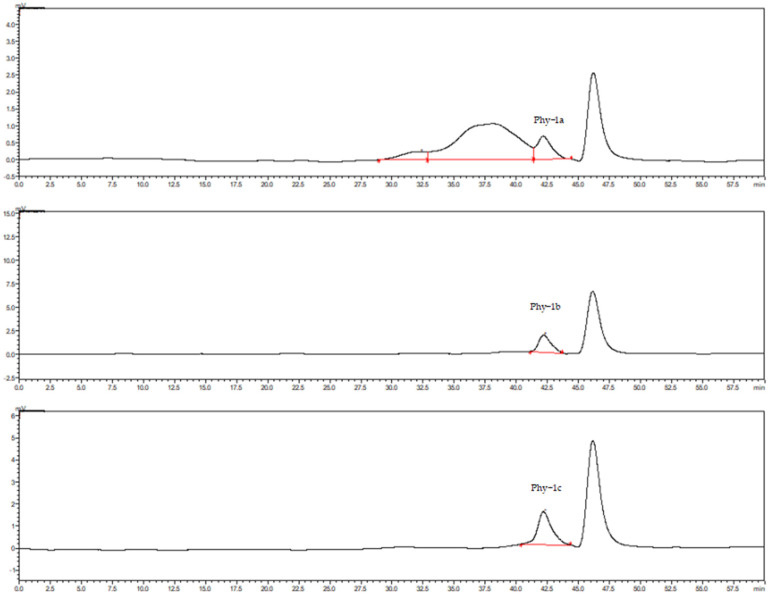
Molecular weight chromatograms of *Physalis alkekengi* L. polysaccharides.

**Figure 3 molecules-30-00949-f003:**
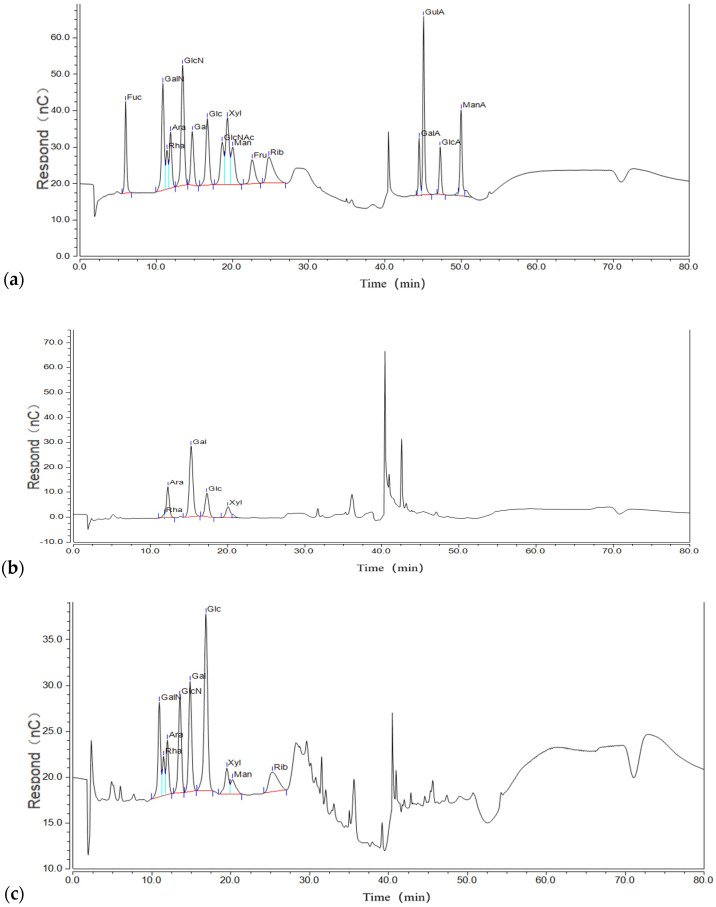
Ion chromatograms of (**a**) mixed monosaccharide standards and *Physalis alkekengi* L. polysaccharides, (**b**) Phy-1b, and (**c**) Phy-1c.

**Figure 4 molecules-30-00949-f004:**
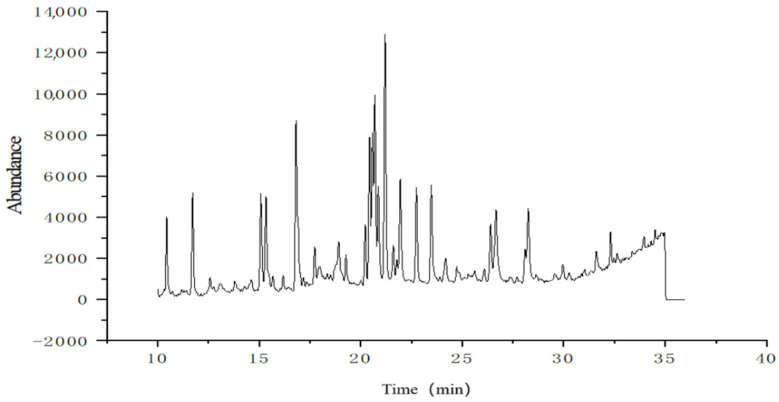
Total ion flow diagram of methylated sugar alcohol acetate of the *Physalis alkekengi* L. polysaccharide Phy-1b.

**Figure 5 molecules-30-00949-f005:**
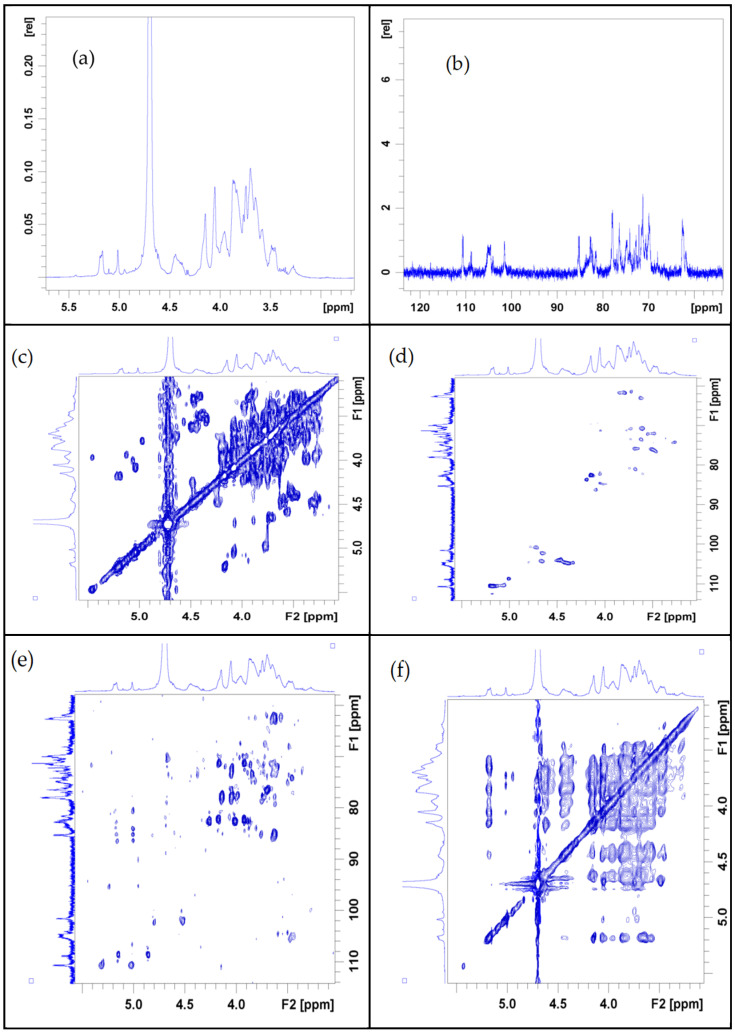
Nuclear magnetic resonance (NMR) spectral profiles of the *Physalis alkekengi* L. polysaccharide Phy-1b. (**a**) 1H-NMR spectrum; (**b**) 13C-NMR spectrum; (**c**) COSY map; (**d**) HSQC map; (**e**) HMBC map; (**f**) NOESY map.

**Figure 6 molecules-30-00949-f006:**
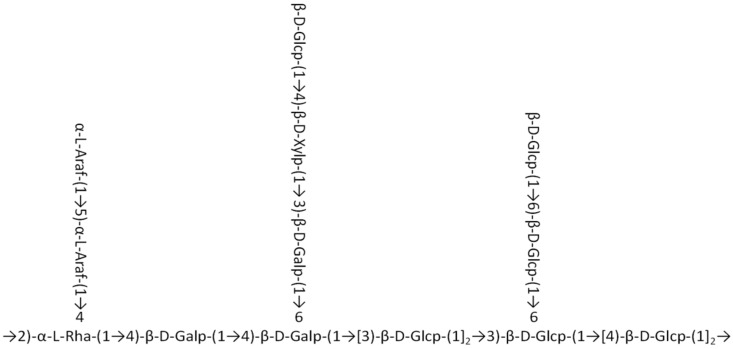
The derived chemical structure formula for the *Physalis alkekengi* L. polysaccharide Phy-1b.

**Table 1 molecules-30-00949-t001:** Monosaccharide composition of the purified fractions of *Physalis alkekengi* L. polysaccharide components.

Monosaccharide	Monosaccharide Molar Ratio of Phy-1b	Monosaccharide Molar Ratio of Phy-1c
fucose	0.00%	0.00%
galactosamine hydrochloride	0.00%	3.90%
rhamnose	3.00%	10.40%
arabinose	19.80%	7.90%
glucosamine hydrochloride	0.00%	5.80%
galactose	47.50%	18.90%
glucose	20.90%	24.70%
xylose	8.80%	4.60%
mannose	0.00%	4.40%
fructose	0.00%	0.00%
ribose	0.00%	19.40%
galactouronic acid	0.00%	0.00%
guluronic acid	0.00%	0.00%
glucuronic acid	0.00%	0.00%
mannuronic acid	0.00%	0.00%

**Table 2 molecules-30-00949-t002:** Methylation analysis of the *Physalis alkekengi* L. polysaccharide Phy-1b and its reduction product determined using gas chromatography–mass spectrometry.

Retention Time	Methylated Sugar Alcohol Acetate	Glycoside Bond Type	Molar Ratio
10.441	2,3,5-Me3-Araf	Araf-(1→	5.2%
15.081	2,3-Me2-Araf	→5)-Araf-(1→	7.1%
15.331	2,3-Me2-Xylp	→4)-Xylp-(1→	7.8%
16.816	2,3,4,6-Me4-Glcp	Glcp-(1→	13.7%
18.917	3-Me1-Rhap	→2,4)-Rhap-(1→	6.3%
20.697	2,4,6-Me3-Glcp	→3)-Glcp-(1→	11.2%
20.864	2,3,6-Me3-Galp	→4)-Galp-(1→	7.0%
21.201	2,3,6-Me3-Glcp	→4)-Glcp-(1→	16.3%
21.952	2,4,6-Me3-Galp	→3)-Galp-(1→	7.4%
22.748	2,3,4-Me3-Glcp	→6-Glcp-(1→	6.0%
26.667	2,4-Me2-Glcp	→3,6)-Glcp-(1→	6.8%
28.251	2,3-Me2-Galp	→4,6)-Galp-(1→	5.3%

**Table 3 molecules-30-00949-t003:** Assignments of 1H-nuclear magnetic resonance (NMR) and 13C-NMR chemical shifts (δ) for the glycosyl residues of the *Physalis alkekengi* L. polysaccharide Phy-1b.

	H1	H2	H3	H4	H5	H6a	H6b
Sugar Residue	C1	C2	C3	C4	C5	C6	
β-L-Araf-(1→	5.17	4.13	3.87	4.05	3.75	3.63	
	110.66	82.81	78.27	85.11	62.64		
→5)-β-L-Araf-(1→	5.01	4.07	3.94	4.19	3.96	3.71	
	108.81	82.18	78.12	83.68	68.27		
→4)-β-D-Xylp-(1→	4.40	3.27	3.50	3.71	36.30	3.46	
	103.03	74.32	76.31	78.72	63.63		
β-D-Glcp-(1→	4.67	3.53	3.69	3.72	3.68	3.83	3.78
	104.27	72.11	75.73	70.95	75.82	61.86	
→3)-β-D-Glcp-(1→	4.68	3.49	3.71	3.47	3.51	3.80	3.85
	104.38	75.96	85.65	72.11	76.28	62.05	
→4)-β-D-Glcp-(1→	4.47	3.27	3.49	3.59	3.47	3.78	3.71
	104.11	74.34	76.72	82.36	76.59	61.64	
→6)-β-D-Glcp-(1→	4.41	3.27	3.47	3.36	3.67	3.61	3.73
	104.85	74.53	72.06	73.36	76.03	70.49	
→3,6)-β-D-Glcp-(1→	4.44	3.30	3.72	3.43	3.63	3.61	3.73
	104.13	74.55	85.91	71.05	76.11	70.49	
→2,4)-α-L-Rhap-(1→	4.71	4.05	3.78	3.68	3.76	1.23	
	100.83	82.04	71.30	80.91	71.36	18.23	
→3)-β-D-Galp-(1→	4.65	3.54	3.74	3.45	3.83	3.46	3.57
	104.22	72.21	80.88	73.17	75.65	63.06	
→4)-β-D-Galp-(1→	4.45	3.27	3.45	3.74	3.83	3.63	3.48
	104.48	74.66	76.62	80.81	75.01	63.05	
→6)-β-D-Galp-(1→	4.35	3.48	3.56	3.72	4.14	3.78	4.04
	104.76	72.16	72.71	73.55	69.79	71.36	
→4,6)-β-D-Galp-(1→	4.43	3.28	3.47	3.61	3.62	3.78	4.04
	104.57	74.34	76.18	83.22	73.71	71.36	

## Data Availability

All data dealing with this study are reported in the paper.

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
