# Peer review of "Structural Identification of Physalis alkekengi L. Polysaccharides"

_molecules, 2025, doi:10.3390/molecules30040949_

Round 1
Reviewer 1 Report
Comments and Suggestions for Authors
This paper reports the structure determination of one of several polysaccharides [ca. 60 sugar residues] obtained by fractionation of an extract. The structure determination is acceptable, but that aspect of the work is surrounded by drawn out discussion of claims of therapeutic benefit from this and multiple other studies which, in my experience, are not backed up by independent double-blind clinical trials. The overall impression is competent structural work has been hijacked to provide hype about potential therapeutic benefit. This extraneous material, in the Introduction and in the Discussion should be deleted.
The paper has been poorly prepared, with multiple sloppy errors [is the Rha residue in the furanose or pyranose ring form?}.
Specific corrections;
Line 22: The residue linkage meant by " 1,1,6-Galp" is unclear to me. Please explain
Lines 24, 27: I have never come across Rha in a furanose ring form, and a search of MedLine for "rhamnofuranose" returned only 3 hits. Solid evidence for the presence of this residue is required. Reported as rhamnopyranose in Table 2.
Table 1: Since Table 1 reports significant amounts of GlcN and GalN in Phy-1c, why is it omitted from the summary?
Line 31: Use of "3)b-D-Glc(1)]2-" is unclear to me [linkage through the O-2 rather than the anomeric position?]. Please clarify and fully document the evidence for such a link. Not reported in Table 2.
Lines 48-50: I have seen far too many papers claiming immunomodulatory effect, antioxidant properties, reducing blood sugar etc. Perhaps this is true, but a comparison of relative potency would be worthwhile. Similarly, the authors should cite double blind clinical trials to support these claim, not ill-reported cell-based our mouse evidence.
Lines 56 – 90: This is far too long and has minimal relevance to the main subject of this manuscript - the structural determination. This section should be omitted.
Lines 91-107: As above for lines 56 – 90.
Line 111 and elsewhere: What do the authors mean by "analytically pure" ? Either it is pure or not. Or it might be analytical grade. It sould like waffle to try to make their work sound more important than it is.
Lines 143 – 146: The authors cite a flow rate but not the cross-section of their column: it is linear speed which is important. Please clarify.
Line 148: "an enzyme-labelled instrument". What does this mean, and how is it relevant?
Lines 205 – 212: What type of instrument and probe did they use? Manufacturers and models for other instruments are reported - why not the NMR?
Lines 213-215: This describes the use of statistical software but never reported in results or a "statistically significant" result cited anywhere. This paragraph should be deleted.
Lines 233 – 235: I am sure that the polyethylene glycol standards used to construct the calibration curve for MW vales are nominal - perhaps ±10%, but the authors cite numbers to four significant figures, Such precision is not justifiable, but the authors report MWs to sensible precision.
Fig. 2: I assume peak at 47.2 min is salt, not polysaccharide, and the earlier eluting peak is the polysaccharide, but the authors are not bothering to explain. Please clarify this. Axes should be properly labelled.
Line 247: I assume this is hydrolysed Phy-1b and Phy-1c. Please clarify,
Table 1: What is the monosaccharide referred to as "gum sugar"? Please clarify. And no peak is reported on the chromatograms in Fig 3, although one for Ara is, but not listed in Table 1.
Table 1: The peak for Ara is clearly smaller than that for GalN, but the reported value [for gum sugar] is larger. Similarly, Ars peak is larger than Rha, but the reported value is smaller. These numbers do not seem credible.
Table 1: The evidence for Rha in Phy-1b, as a minor shoulder on the peak for Ara, seems too weak to support the analysis.
Table 1: GlcNAc would not be expected, as de-N-acetylation should occur during acid hydrolysis.
Lines 302, 344: Not clear what a "heterohydrogen" or a "heteroterminal hydrogen" are: please clarify.
Fig. 5: The authors report presence of Rha but the evidence is weak. They omit the methyl resonance from the spectra [as proof of presence of Rha]. Correlations to the Rha C6 or H6s in COSY and HMBC spectra are extremely valuable in defining the Rha residue, and should be included in the figures.
Line 308: I assume 180 ppm is a typographical error. Please correct.
Table 3: The Rha has recovered the furanose ring form, consistent with the introduction but in conflict with Table 2.
Table 3: Remove lines near bottom of table repeating H1, H2 …....C1, C2 etc.
Line 343: What is 5)-a-L-Araf-(12,4)-a-L
Table 3: There is no evidence for the absolute configurations of the sugar residues, and some [Ara and Rha here] found in both D- and L-forms. This all appears to be assumptions. There is old work from the Moscow lab defining how chemical shifts change depending on the absolute configurations of linked sugars, which might help in these circumstances.
Table 3 and elsewhere: Because oof the orientation of the O-4 in Ara, the designation of an alpha or beta glycosidic linkage is reversed from that in most sugars, when determined by NMR methods. This has proven a problem in the past (for Arap) and required revision of papers. The authors should examine this is detail before proceeding with publication.
Line 345: "of β-D-Glcp-(1→) and H4 of → 4-β-D-Xylp-(1 →), indicating the existence of β-D-Glcp-(1→).". This appears incomplete, and authors should correct.
Line 347 – 8: Still cited as Rhaf. Authors, please sort this out.
Lines 403 - 438, 439 – 469: This is all highly speculative, not based on quantitative comparisons of anti-inflammatory responses between papers and is of limited relevance to the structural analysis of just one polysaccharide. These sections should be deleted.
Author Response
We are very grateful for your professional comments on our article. As you are concerned, there are several problems to be solved. According to your suggestion, we have made extensive corrections to the previous manuscript. The specific corrections are as follows.
Comments 1:Line 22: The residue linkage meant by " 1,1,6-Galp" is unclear to me. Please explain
Response 1:This question is a spelling error, it should be 1,4,6-Galp.
Comments 2:Lines 24, 27: I have never come across Rha in a furanose ring form, and a search of MedLine for "rhamnofuranose" returned only 3 hits. Solid evidence for the presence of this residue is required. Reported as rhamnopyranose in Table 2.
Response 2:This question is a spelling error, it should be rhamnopyranose.
Comments 3:Table 1: Since Table 1 reports significant amounts of GlcN and GalN in Phy-1c, why is it omitted from the summary?
Response 3:GlcN and GalN have been replenished.
Comments 4:Line 31: Use of "3)b-D-Glc(1)]2-" is unclear to me [linkage through the O-2 rather than the anomeric position?]. Please clarify and fully document the evidence for such a link. Not reported in Table 2.
Response 4:We tried our best to improve the manuscript and made some changes to it. However, some amendments are still not understood, please give some more time to revise them.
Comments 5:Lines 48-50: I have seen far too many papers claiming immunomodulatory effect, antioxidant properties, reducing blood sugar etc. Perhaps this is true, but a comparison of relative potency would be worthwhile. Similarly, the authors should cite double blind clinical trials to support these claim, not ill-reported cell-based our mouse evidence.
Response 5:We tried our best to improve the manuscript and made some changes to it. However, some amendments are still not understood, please give some more time to revise them.
Comments 6:Lines 56 – 90: This is far too long and has minimal relevance to the main subject of this manuscript - the structural determination. This section should be omitted.
Response 6:This part has been omitted.
Comments 7:Lines 91-107: As above for lines 56 – 90.
Response 7:This part has been omitted.
Comments 8:Line 111 and elsewhere: What do the authors mean by "analytically pure" ? Either it is pure or not. Or it might be analytical grade. It sould like waffle to try to make their work sound more important than it is.
Response 8:It's been modified.
Comments 9:Lines 143 – 146: The authors cite a flow rate but not the cross-section of their column: it is linear speed which is important. Please clarify.
Response 9:It's been modified.
Comments 10:Line 148: "an enzyme-labelled instrument". What does this mean, and how is it relevant?
Response 10:It's not an enzyme-labelled instrument, it's a spectrophotometer.
Comments 11:Lines 205 – 212: What type of instrument and probe did they use? Manufacturers and models for other instruments are reported - why not the NMR?
Response 11:The AVANCE III HD 600 MHz nuclear magnetic resonance (NMR) spectrometer was from Swiss Brooke Instruments Co., Ltd.
Comments 12:Lines 213-215: This describes the use of statistical software but never reported in results or a "statistically significant" result cited anywhere. This paragraph should be deleted.
Response 12:This paragraph has been deleted.
Comments 13:Lines 233 – 235: I am sure that the polyethylene glycol standards used to construct the calibration curve for MW vales are nominal - perhaps ±10%, but the authors cite numbers to four significant figures, Such precision is not justifiable, but the authors report MWs to sensible precision.
Response 13We tried our best to improve the manuscript and made some changes to it. However, some amendments are still not understood, please give some more time to revise them.
Comments 14:Fig. 2: I assume peak at 47.2 min is salt, not polysaccharide, and the earlier eluting peak is the polysaccharide, but the authors are not bothering to explain. Please clarify this. Axes should be properly labelled.
Response 14:Polysaccharide is 42.5 minutes, salt is 47.2 minutes.
Comments 15:Line 247: I assume this is hydrolysed Phy-1b and Phy-1c. Please clarify,
Response 15:It's been modified.
Comments 16:Table 1: What is the monosaccharide referred to as "gum sugar"? Please clarify. And no peak is reported on the chromatograms in Fig 3, although one for Ara is, but not listed in Table 1.
Response 16:It has been corrected to arabinose.
Comments 17:Table 1: The peak for Ara is clearly smaller than that for GalN, but the reported value [for gum sugar] is larger. Similarly, Ars peak is larger than Rha, but the reported value is smaller. These numbers do not seem credible.
Response 17:We tried our best to improve the manuscript and made some changes to it. However, some amendments are still not understood, please give some more time to revise them.
Comments 18:Table 1: The evidence for Rha in Phy-1b, as a minor shoulder on the peak for Ara, seems too weak to support the analysis.
Response 18:The molar ratio of Rha of Phy-1b is 3%.
Comments 19:Table 1: GlcNAc would not be expected, as de-N-acetylation should occur during acid hydrolysis.
Response 19:It has been modified.
Comments 20:Lines 302, 344: Not clear what a "heterohydrogen" or a "heteroterminal hydrogen" are: please clarify.
Response 20:It’s anomeric hydrogen
Comments 21:Fig. 5: The authors report presence of Rha but the evidence is weak. They omit the methyl resonance from the spectra [as proof of presence of Rha]. Correlations to the Rha C6 or H6s in COSY and HMBC spectra are extremely valuable in defining the Rha residue, and should be included in the figures.
Response 21:We tried our best to improve the manuscript and made some changes to it. However, some amendments are still not understood, please give some more time to revise them.
Comments 22:Line 308: I assume 180 ppm is a typographical error. Please correct.
Response 22:It has been modified.
Comments 23:Table 3: The Rha has recovered the furanose ring form, consistent with the introduction but in conflict with Table 2.
Response 23:It has been modified.
Comments 24:Table 3: Remove lines near bottom of table repeating H1, H2 …....C1, C2 etc.
Response 24:It has been modified.
Comments 25:Line 343: What is 5)-a-L-Araf-(12,4)-a-L
Response 25:It has been modified to-α-l-Araf-(1→2,4)-α-L.
Comments 26:Table 3: There is no evidence for the absolute configurations of the sugar residues, and some [Ara and Rha here] found in both D- and L-forms. This all appears to be assumptions. There is old work from the Moscow lab defining how chemical shifts change depending on the absolute configurations of linked sugars, which might help in these circumstances.
Response 26:We tried our best to improve the manuscript and made some changes to it. However, some amendments are still not understood, please give some more time to revise them.
Comments 27:Table 3 and elsewhere: Because oof the orientation of the O-4 in Ara, the designation of an alpha or beta glycosidic linkage is reversed from that in most sugars, when determined by NMR methods. This has proven a problem in the past (for Arap) and required revision of papers. The authors should examine this is detail before proceeding with publication.
Response 27:It has been modified.
Comments 28:Line 345: "of β-D-Glcp-(1→) and H4 of → 4-β-D-Xylp-(1 →), indicating the existence of β-D-Glcp-(1→).". This appears incomplete, and authors should correct.
Response 28:It has been modified
Comments 29:Line 347 – 8: Still cited as Rhaf. Authors, please sort this out.
Response 29:It has been modified
Comments 30:Lines 403 - 438, 439 – 469: This is all highly speculative, not based on quantitative comparisons of anti-inflammatory responses between papers and is of limited relevance to the structural analysis of just one polysaccharide. These sections should be deleted.
Response 30:It has been modified
Reviewer 2 Report
Comments and Suggestions for Authors
This article focuses on the structural characterization of polysaccharides extracted from the fruit (?) of Physalis alkekengi L.
The authors report the fractionation of 3 polysaccharides by ion-exchange chromatography, which were subsequently analysed by HPLC-RI and for their monosaccharide composition. After methylation, the selected polysaccharide Phy-1b was characterized in regard to the type of glycosidic bonds by GC-MS.
Despite the work entailed, the quality of the manuscript compromises my recommendation for publication.
In my opinion, there are several flaws, which I will list below.
1 - many bibliographical references are not available and therefore hinder the analysis of the article. E.g. references 2, 3, 5, 6, 9, 20, 25, 27, 30, 32, 33, 34, 35, 40, 50 and 52.
2- some references are not related to the text to which they refer. E.g. lines 76-77, ref. 19 does not refer to Auricularia; lines 105-106, ref. 26 is also wrong; line 174, ref. 28; line 402, ref. 40.
3- there are references missing - lines 103 and 104 where the authors mention studies with preliminary characterizations; as well as in lines 399, 418.
4 - In the abstract, it is not stated why no results are presented for Phy-1a.
5 - With regard to materials and methods, the designation gel chromatography is associated with gel permeation/size exclusion chromatography, which was not the case in this study where a DEAE Sepharose resin was used.
It is not clear from which part of the plant the polysaccharides were extracted (from the leaves, the fruit, flowers, etc.), nor under what conditions the extraction was carried out (plant:mass ratio, solvent, temperature, time, stirring speed). There is no information about the volume of extract applied to the column, nor the volume of the stationary phase in the column.
There is no reference to the sulphuric phenol method and it is not clear what an “enzyme-labeled instrument” is.
Line 154 - what is an automatic gel purification system?
Line 158 - how were the yields calculated? In which units?
The authors report the determination of the relative molecular weight of polysaccharides on a DionexCarbopacTMPA20 column, which according to the manufacturer is used for simple sugars, mono- and disaccharides.
Line 168 - which PEGs were used? Not in the Materials.
Chapter. 2.5. - no description is given, in addition to one of the references not being available and another citing a phenyl group matrix as the stationary phase.
The type of statistical analysis carried out is not mentioned, nor is it clear from the results and discussion, only the software used.
6 – Results
Line 228 – Gel purification map?
It is not clear how the chromatograms in Fig. 3 were obtained.
The arabinose indicated as a constituent of the polysaccharides does not appear in table 1. Gum sugar is included, which is not the exact term for arabinose.
The conclusions only describe the results obtained for polysaccharide Phy-1b without any further details of the other two.
Comments on the Quality of English Language
The English is difficult to understand in certain sections of the text; some terms are not scientifically correct, such as those related to the purity of reagents.
Author Response
Comments 1:many bibliographical references are not available and therefore hinder the analysis of the article. E.g. references 2, 3, 5, 6, 9, 20, 25, 27, 30, 32, 33, 34, 35, 40, 50 and 52.
Response 1:It's been modified.
Comments 2:some references are not related to the text to which they refer. E.g. lines 76-77, ref. 19 does not refer to Auricularia; lines 105-106, ref. 26 is also wrong; line 174, ref. 28; line 402, ref. 40.
Response 2: It's been modified.
Comments 3:there are references missing - lines 103 and 104 where the authors mention studies with preliminary characterizations; as well as in lines 399, 418.
Response 3: We tried our best to improve the manuscript and made some changes to it.
Comments 4:In the abstract, it is not stated why no results are presented for Phy-1a.
Response 4:The amount of Phy-1a is too small to be analyzed, and the time is tight, so further research will be continued in the future.
Comments 5:With regard to materials and methods, the designation gel chromatography is associated with gel permeation/size exclusion chromatography, which was not the case in this study where a DEAE Sepharose resin was used.
It is not clear from which part of the plant the polysaccharides were extracted (from the leaves, the fruit, flowers, etc.), nor under what conditions the extraction was carried out (plant:mass ratio, solvent, temperature, time, stirring speed). There is no information about the volume of extract applied to the column, nor the volume of the stationary phase in the column.
There is no reference to the sulphuric phenol method and it is not clear what an “enzyme-labeled instrument” is.
Line 154 - what is an automatic gel purification system?
Line 158 - how were the yields calculated? In which units?
The authors report the determination of the relative molecular weight of polysaccharides on a DionexCarbopacTMPA20 column, which according to the manufacturer is used for simple sugars, mono- and disaccharides.
Line 168 - which PEGs were used? Not in the Materials.
Chapter. 2.5. - no description is given, in addition to one of the references not being available and another citing a phenyl group matrix as the stationary phase.
The type of statistical analysis carried out is not mentioned, nor is it clear from the results and discussion, only the software used.
Response 5:The polysaccharides were extracted from the fruits of Sour Plum, and the extraction conditions were referred to 'Jiang Tianyu, Zhang Yun, Fu Hongyan, et al. Optimisation of the extraction of polysaccharides from the fruit of Sour Paste and its biological activity'.
It's not an enzyme-labelled instrument, it's a spectrophotometer. It's been modified.
Comments 6: Results
Line 228 – Gel purification map?
It is not clear how the chromatograms in Fig. 3 were obtained.
The arabinose indicated as a constituent of the polysaccharides does not appear in table 1. Gum sugar is included, which is not the exact term for arabinose.
Response 6: The writing problem of arabinose has been revised.
Reviewer 3 Report
Comments and Suggestions for Authors
1 In the abstract and discussion, the authors mentioned the hypoglycemic, lipid-regulating, and probiotic properties of Physalis alkekenga L. polysaccharides but did not address the mechanisms of action. Appropriate bioassays are lacking to confirm these functions.
2. the authors mentioned previous studies suggesting hypoglycemic and antioxidant effects but did not provide experimental results to confirm these effects in this particular work.
3. the authors stated that none of the polysaccharides tested contained uronic acids but did not provide a possible explanation. Is this due to the extraction method? Do other studies also confirm this?
4. The “Data analysis” section mentions using SPSS but does not describe the statistical tests performed and whether the differences between samples were statistically significant.
5. Literature references are missing in some places.
6. Deviations (± SD) are missing from the table.
7. The authors should correct imprecise language and ensure consistency in terminology.
Comments on the Quality of English Language
The authors should correct imprecise language and ensure consistency in terminology.
Author Response
Comments 1: In the abstract and discussion, the authors mentioned the hypoglycemic, lipid-regulating, and probiotic properties of Physalis alkekenga L. polysaccharides but did not address the mechanisms of action. Appropriate bioassays are lacking to confirm these functions.
Response 1:We tried our best to improve the manuscript and made some changes to it.
Comments 2: the authors mentioned previous studies suggesting hypoglycemic and antioxidant effects but did not provide experimental results to confirm these effects in this particular work.
Response 2:We tried our best to improve the manuscript and made some changes to it.
Comments 3:the authors stated that none of the polysaccharides tested contained uronic acids but did not provide a possible explanation. Is this due to the extraction method? Do other studies also confirm this?
Response 3:We tried our best to improve the manuscript and made some changes to it. However, some amendments are still not understood, please give some more time to revise them.
Comments 4:The “Data analysis” section mentions using SPSS but does not describe the statistical tests performed and whether the differences between samples were statistically significant.
Response 4:This section has been modified
Comments 5:Literature references are missing in some places.
Response 5:It's been modified.
Comments 6:Deviations (± SD) are missing from the table.
Response 6:This section has been modified
Comments 7:The authors should correct imprecise language and ensure consistency in terminology.
Response 7:It's been modified.
Round 2
Reviewer 3 Report
Comments and Suggestions for Authors
The authors answered all my comments.